# *N*-Octyl Caffeamide, a Caffeic Acid Amide Derivative, Prevents Progression of Diabetes and Hepatic Steatosis in High-Fat Diet Induced Obese Mice

**DOI:** 10.3390/ijms23168948

**Published:** 2022-08-11

**Authors:** Miao-Yi Wu, Chia-Chu Liu, Su-Chu Lee, Yueh-Hsiung Kuo, Tusty-Jiuan Hsieh

**Affiliations:** 1Graduate Institute of Medicine, College of Medicine, Kaohsiung Medical University, Kaohsiung 807378, Taiwan; 2Research Center for Environmental Medicine, Kaohsiung Medical University, Kaohsiung 807378, Taiwan; 3School of Medicine, College of Medicine, Kaohsiung Medical University, Kaohsiung 807378, Taiwan; 4Department of Urology, Kaohsiung Medical University Hospital, Kaohsiung Medical University, Kaohsiung 807378, Taiwan; 5Department of Urology, Pingtung Hospital, Ministry of Health and Welfare, Pingtung City 900027, Taiwan; 6Division of Nephrology, Department of Internal Medicine, Kaohsiung Medical University Hospital, Kaohsiung Medical University, Kaohsiung 807378, Taiwan; 7Department of Chinese Pharmaceutical Sciences and Chinese Medicine Resources, China Medical University, Taichung 406040, Taiwan; 8Department of Biotechnology, Asia University, Taichung 413305, Taiwan; 9Chinese Medicine Research Center, China Medical University, Taichung 406040, Taiwan; 10Department of Marine Biotechnology and Resources, College of Marine Sciences, National Sun Yat-sen University, Kaohsiung 804201, Taiwan

**Keywords:** AMPK, PTP1B, obesity, lipid droplet, perilipin, Fsp27

## Abstract

The underlying pathological mechanisms of diabetes are complicated and varied in diabetic patients, which may lead to the current medications often failing to maintain glycemic control in the long term. Thus, the discovery of diverse new compounds for developing medicines to treat diabetes and its complications are urgently needed. Polyphenols are metabolites of plants and have been employed in the prevention and treatment of a variety of diseases. Caffeic acid phenethyl ester (CAPE) is a category of compounds structurally similar to polyphenols. In this study, we aimed to investigate the antidiabetic activity and potential molecular mechanisms of a novel synthetic CAPE derivative *N*-octyl caffeamide (36M) using high-fat (HF) diet induced obese mouse models. Our results demonstrate that 36M prevented the progression of diabetes in the HF diet fed obese mice via increasing phosphorylation of adenosine monophosphate-activated protein kinase (AMPK) and inhibiting expression of protein tyrosine phosphatase 1B (PTP1B). We also found that 36M could prevent hepatic lipid storage in the HF diet fed mice via inhibition of fatty acid synthase and lipid droplet proteins, including perilipins and Fsp27. In conclusion, 36M is a potential candidate compound that can be developed as AMPK inhibitor and PTP1B inhibitor for treating diabetes and hepatic steatosis.

## 1. Introduction

Insulin resistance and type 2 diabetes mellitus (T2DM) are growing health issues and life-threatening events worldwide [1,2]. Diabetic complications, such as neuropathy, nephropathy, retinopathy, cardiovascular diseases, peripheral vascular diseases, sacorpenia, stroke, amputations, renal failure, and blindness, result in increased disability, reduced life expectancy, and enormous healthcare costs [1,3]. The International Diabetes Federation (IDF) Diabetes Atlas reported that the number of diabetic patients will increase at an exponential rate from 537 to 783 million by the year 2045 [4]. The underlying pathological mechanisms of diabetes are complicated and varied in diabetic patients, which may lead to the current medications often failing to maintain glycemic control in the long term and the patients eventually suffering from problems such as short duration, risk of hypoglycemia and steady weight gain, fluid retention, heart failure, fractures, efficacy, cost, and comorbidities [5]. Thus, the discovery of diverse new compounds for developing medicines to treat diabetes and its complications is urgently needed.

Excess body weight has been recognized as a risk factor for T2DM in which approximately 90% of affected patients are overweight or obese [6,7]. Unhealthy dietary intake, particularly a high-fat (HF) diet, plays a critical role to cause imbalance of glucose and lipid metabolism, which gradually impair the function of major metabolic organs leading to dyslipidemia, hyperinsulinemia, progressive insulin resistance, late pancreatic-cell failure, and T2DM [7,8]. The overwhelming role of HF diet in the formation of metabolic disorders, such as systemic insulin resistance and T2DM, is demonstrated by a variety of experiments performed on humans, rodents, and cell culture models [8,9,10,11,12].

Adenosine monophosphate-activated protein kinase (AMPK) is a key player in the pathways that mediate blood glucose entering peripheral tissues, such as skeletal muscle and inhibiting gluconeogenesis in the liver [13,14,15]. The American Diabetes Association (ADA) suggests that metformin, an AMPK activator, is the optimal first-line drug for treating hyperglycemia [16]. Hence, AMPK is always a favorite drug target for metabolic diseases. Many natural products and their synthetic derivatives have been claimed to activate the AMPK pathway and show beneficial effects in metabolic disorders such as DM [5]. Accumulated evidence demonstrates that polyphenols (i.e., flavanones, flavonols, chalcones, and caffeic acids) modulate AMPK activation and become favorable therapeutic strategies to treat or prevent obesity and diabetes [5,17,18,19].

Polyphenols are metabolites of plants and have been employed in the prevention and treatment of a variety of diseases, including cancers, cardiovascular diseases, diabetes, obesity, osteoporosis, and neurodegenerative diseases [5,20,21]. Many naturally occurring compounds, such as resveratrol, quercetin, theaflavin, berberine, curcumin, and caffeic acid phenethyl ester (CAPE), are structurally similar to polyphenols [22]. CAPE is one of the major components in honeybee propolis and has been reported to possess antioxidant, anti-inflammatory, proapoptotic, antiviral, and immunomodulatory bioactivities [22].

Kuo et al. developed a series of CAPE derivatives and found some of them carried bioactivities to prevent renal fibrosis, cardiac dysfunction, neuron inflammation, and hepatic glucose production [23,24,25,26,27]. CAPE and the derivative KS370G (36) and 36-13 have been shown to possess hypoglycemic activities in streptozotocin (STZ)-induced and/or high-fat/fructose diet-induced diabetic mouse models [22,27,28]. In this study, we aimed to investigate the antidiabetic activity and potential molecular mechanisms of another novel synthetic CAPE derivative 36M (Figure 1A) that is the same series derivatives of KS370G in high-fat diet induced obese mice.

## 2. Results

### 2.1. N-Octyl Caffeamide (36M) Increases Glucose Consumption in 3T3-L1 Adipocytes and C2C12 Myotubes

To evaluate the potential bioactivity of 36M on anti-diabetes, we first compared its glucose-lowering effect with rosiglitazone and 36 in 3T3-L1 adipocytes. As shown in Figure 1B, rosiglitazone, 36 and 36M could all increase consumption of the glucose in the medium after administered for 24 h. Compared at the 25 μg/mL concentration level, 36M demonstrated the best efficacy to increase glucose consumption in the adipocytes (Figure 1B). We also evaluated the glucose-lowering effect of 36M in C2C12 myotubes and showed that 25 μg/mL of 36M significantly increased glucose consumption and the effect was similar to that of 0.32 μM insulin (Figure 1C).

### 2.2. 36M Is an AMPK Activator and a PTP1B Inhibitor

To explore the underlying mechanism of 36M on its anti-diabetic effect, we observed the influence of 36 M on AMPK and protein tyrosine phosphatase 1B (PTP1B) in C2C12 myotubes. As shown in Figure 2, 36M significantly increased AMPK phosphorylation and decreased PTP1B protein level in a dose-dependent manner, indicating that 36M is an AMPK activator and a PTP1B inhibitor.

### 2.3. 36M Prevents Progression of Diabetes in HF-Induced Obese Mice

To confirm the anti-diabetic effect of 36M, we first tested its efficacy using a prevention model in which 36M was administered after HF feeding for two weeks (Figure 3A). In this model, 36M was administered before the fasting blood glucose was elevated in the HF-fed mice (Figure 3B). After administered 36M for 10 weeks, we found the body weight of the HF-fed mice was significantly less than that of the HF-fed mice without administered 36M (Figure 3C). The OGTT result demonstrated that 36M could maintain the glucose tolerance of the HF-fed mice as that of the control mice (Figure 3D,E). The efficacy of 36M was similar to that of pioglitazone (Figure 3D,E). At the end point of the experiment, 36M significantly prevented the elevation of fasting and postprandial (post cibum) blood glucose (Figure 3F,G). The efficacy of 36M on preventing occurrence of diabetes was similar to that of pioglitazone (Figure 3F,G). However, both 36M and pioglitazone could not prevent the increase in plasma total cholesterol, triglyceride, and ALT (Figure 3H–J).

In the second animal model (Figure 4A), we treated the obese mice with 36M or pioglitazone after 8 weeks of HF feeding. In this treatment model, 36M improved glucose intolerance induced by HF feeding (Figure 4B,C). At the end of the experiment, similar to pioglitazone, 36M significantly inhibited the progression of diabetes and the increase in body weight (Figure 4D–F). In the treatment model, 36M also could not significantly decrease plasma total cholesterol, triglyceride, and ALT (Figure 4G–I).

### 2.4. 36M Increases AMPK Activation and GLUT4 Expression in Skeletal Muscle of HF Fed Mice

In the prevention model, AMPK phosphorylation in skeletal muscle was significantly higher in HF-fed mice than that in chow-diet fed mice, indicating that AMPK activity remained to be preserved in HF-fed mice (Figure 5A,B). 36M was demonstrated to further increase AMPK phosphorylation in the skeletal muscle of the HF-fed mice (Figure 5A,B). In the treatment model, AMPK activity was impaired in the skeletal muscle of HF fed mice but the activity was preserved by 36M treatment (Figure 5D,E). In both models, we found that protein expression of glucose transporter 4 (GLUT4) was remarkably increased in the skeletal muscle of HF-fed mice treated with 36M (Figure 5A,C,D,F).

### 2.5. 36M Preserves Insulin Signaling in Skeletal Muscle of HF-Fed Mice

As shown in Figure 6A,B, in the prevention model, there was no significant difference of the protein level of insulin receptor subunit β (IRβ) among the three groups. However, phosphorylation of IRβ was significantly increased in the skeletal muscle of HF and HF + 36M groups (Figure 6A,B). It is noteworthy that the administration of 36M resulted in remarkable increase in IRβ phosphorylation than that in the mice only fed with HF (Figure 6A,B). Phosphorylation of protein kinase B (AKT) was also increased in the skeletal muscle of the HF-fed mice treated with 36M (Figure 6A,C). As shown in Figure 6F,G, in the treatment model, we observed that 36M increased IRβ phosphorylation and IRβ protein expression. Protein expression and phosphorylation of AKT were also upregulated by 36M treatment (Figure 6F,H).

In the prevention model, phosphorylation levels of glycogen synthase kinase-3 (GSK3β) and glycogen synthase (GS) in skeletal muscle were higher in the HF + 36M group than that in the HF group, but the difference did not reach statistical significance (Figure 6A,D,E). In the treatment model, even though GSK3β phosphorylation in skeletal muscle was significantly increased by 36M treatment, GS phosphorylation was not changed by 36M (Figure 6F–J).

### 2.6. 36M Prevents Lipid Droplet Accumulation and Activates AMPK in Liver of HF-Fed Mice

After being fed with a HF diet, abundant lipid droplets were accumulated in the livers of the HF group (Figure 7A,B). In contrast, administration of 36M to the HF-fed mice significantly prevented accumulation of lipid droplets in the livers (Figure 7A,B). In addition, the phosphorylation of AMPK and acetyl-CoA carboxylase 1 (ACC1) was significantly increased by 36M treatment in both prevention model (Figure 8A–C) and treatment model (Figure 8D–F).

### 2.7. 36M Decreases Fatty Acid Synthase and Lipid Droplet Proteins in Liver of HF-Fed Mice

As shown in Figure 9A–C,E, protein levels of sterol regulatory element-binding protein 1c (SREBP1c), peroxisome proliferator-activated receptor gamma (PPARγ), and perilipin 1 showed decreased trends in the liver of the HF-fed mice with 36M treatment compared to that of the HF-fed mice. However, the decreased trends did not reach statistical significance. Protein levels of fatty acid synthase (FASN), perilipin 2, perilipin 5, and Fsp27 were significantly decreased in the liver of HF + 36M group than that of HF group (Figure 9A,D,F–H).

### 2.8. 36M Inhibits PTP1B Expression in Skeletal Muscle and Liver of HF-Fed Mice

We also observed the protein expression of PTP1B in skeletal muscle and liver of the mice. As shown in Figure 10A–D, PTP1B protein expression was significantly inhibited by 36M treatment, both in the prevention and the treatment models.

## 3. Discussion

Many natural compounds, such as ascorbic acid and polyphenols, found in fruits and vegetables have been demonstrated to reduce oxidative stress and inflammation and have been suggested to have beneficial effects that against various diseases [5,29]. CAPE is structurally similar to polyphenols and has been reported to possess many bioactivities [22]. We developed and reported a series of CAPE derivatives that could prevent renal fibrosis, cardiac dysfunction, neuron inflammation, and hepatic glucose production [22,23,24,25,26,27], especially the derivative KS370G (36) and 36-13 carrying hypoglycemic activities in several diabetic animal models [22,27,28]. In this study, we explored the antidiabetic activity targeting T2DM and potential molecular mechanisms of the novel synthetic CAPE derivative 36M. We first tested its potential glucose lowering effect in 3T3-L1 adipocytes and C2C12 myotubes. AMPK and PTP1B are the two favorite targets of anti-diabetic drug discovery [18,30]. We also tested whether 36M could influence AMPK and PTP1B. In the in vitro models, the results showed that 36M existed excellent activity on lowering glucose, activating AMPK, and inhibiting PTP1B. These results indicated that 36M could be an ideal candidate compound to develop drugs against T2DM. Thus, we further confirmed the antidiabetic potential of 36M using high-fat diet induced obese mouse models. Our results demonstrate that 36M prevents the progression of diabetes and hepatic steatosis, at least in part, via AMPK activation.

AMPK is a key master that regulates many cellular functions, such as lipid, glucose, and protein metabolism; cellular growth; and mitochondrial biogenesis and autophagy [31,32]. There is growing interest in developing AMPK activators because AMPK has been suggested to have beneficial clinical effects in various metabolic and degenerative diseases, aging, diabetes, cancer, and viral infection [17,18,32,33]. Various direct and indirect AMPK activators have been reported to maintain the homeostasis of glucose and lipid metabolism and may improve insulin sensitivity [18]. Thus, AMPK activators may emerge as a novel therapy in diabetes and its associated complications. Habegger et al. reported that AMPK could enhance insulin-stimulated GLUT4 regulation via lowering membrane cholesterol and this AMPK-mediated process could protect against hyperinsulinemia-induced insulin resistance [34]. Skeletal muscle is the major organ for energy expenditure with glucose uptake. Transcription of GLUT4 is regulated by AMPK activation [35]. Thus, we observed AMPK phosphorylation and GLUT4 protein expression in skeletal muscle. Specific overexpression of GLUT4 in skeletal muscle ameliorated insulin resistance associated with obesity and diabetes [35,36,37,38]. In human skeletal muscle, AMPK was found to regulate the GLUT4 gene expression by causing phosphorylation of histone deacetylase 5 (HDAC5) on serines^259^ and serines^498^ which led to dissociation of HDAC5 from myocyte enhancer factor 2 (MEF2, a pivotal regulator of GLUT4 expression) and then allowed MEF2-mediated transcription to proceed through a region containing the MEF2 binding site on the GLUT4 gene promoter [35]. In our study, we found that 36M processed glucose lowering effect in the C2C12 myotubes and prevented progression of hyperglycemia in the HF induced obese mice. The effect may result from activation of AMPK that leads to overexpression of GLUT4 in skeletal muscle of the obese mice. IRβ and AKT are downstream of insulin signaling and are responsible for the regulation of glucose uptake by mediating insulin-induced translocation of GLUT4 to the cell surface [39]. We found that 36M preserved glucose tolerance in the obese mice which may be due to maintaining or increasing the sensitivity of insulin signaling by stimulating phosphorylation of IRβ and AKT in skeletal muscle.

In response to insulin, AKT activation results in GSK3 phosphorylation [40,41]. GSK3 inhibits GS via phosphorylation and has important regulatory functions in glucose metabolism, insulin activity, and energy homeostasis. Insulin induces GSK3 phosphorylation and inactivation, which allows GS activation and glucose deposition as glycogen [41]. In addition, GSK3β has inhibitory effects on skeletal muscle via blockage of protein synthesis. Increase in AKT phosphorylation induces GSK3β phosphorylation and results in GSK3β inactivation which enhances translation initiation and elongation and, thus, protein synthesis [42]. However, 36M did not have significant effects on promotion of glycogen synthesis and protein production since phosphorylation of GSK3β and GS did not change in skeletal muscle of the HF fed mice.

Liver is the core organ that regulates glucose and lipid metabolism. Nonalcoholic fatty liver disease (NAFLD) is one of the most common liver diseases and encompasses a wide range of downstream pathologies, including simple steatosis, steatohepatitis, advanced fibrosis, and cirrhosis [43,44,45]. After consumption of a high carbohydrate diet, patients with insulin resistance present lower glucose uptake and glycogen synthesis in skeletal muscle, leading to a doubling of both liver triglyceride levels and hepatic de novo lipogenesis [46,47]. This phenomenon suggests that insulin resistance in the skeletal muscle shifts post-prandial energy storage from muscle glycogen to hepatic lipid storage [47]. AMPK is a common characteristic of many proposed NAFLD treatment options [48]. In the liver, liver kinase B1 (LKB1), a serine/threonine kinase, is the major upstream kinase that regulates AMPK activity via activating phosphorylation of the α-subunit at Thr^172^ residue [48]. AMPK activation may reduce NAFLD by suppressing de novo lipogenesis and increasing fatty acid oxidation in liver [48,49]. Phosphorylation of ACC1 at Ser^79^ and ACC2 at Ser^221^ by AMPK activation blocks ACC dimerization that causes a reduction in ACC activity, leading to decrease in malonyl-CoA, inhibition of de novo lipogenesis. and increase in mitochondrial fatty acid oxidation [48,49]. Sterol regulatory element-binding proteins (SREBPs) are the major transcriptional factors of many lipogenic genes, including FASN [50]. Peroxisome proliferator-activated receptors (PPARs) play key roles in the transcriptional regulation of glucose and lipid metabolism [51]. Thus, we observed the expression of these proteins to clarify the molecular mechanisms of 36M on the prevention of NAFLD. AMPK downregulates lipogenic gene expression by interacting with SREBP-1c [48]. FASN catalyzes the last step in fatty acid biosynthesis and may play as a major determinant of the maximal hepatic capacity to generate fatty acids by de novo lipogenesis [50]. Hepatic FASN expression was reported to be significantly increased in vivo in a murine model of hepatic steatosis and correlated with the degree of hepatic steatosis in human NAFLD livers [50]. SREBP-1c was reported to be induced by feeding a high carbohydrate diet that subsequently promoted hepatic FASN expression in rats [50]. Our results demonstrate that 36M inhibited hepatic steatosis and increased phosphorylation of AMPK and ACC1. Furthermore, 36M partially reduced protein levels of SREBP-1c and FASN. These data suggest that 36M may inhibit de novo lipogenesis in liver of the HF induced obese mice.

Lipid droplets (LDs) are fat-storage organelles existing in most cell types and serve as readily accessible reservoirs of high-energy substrates to be used for β-oxidation within mitochondria [52,53,54]. LDs are coated by several proteins, including perilipins and Fsp27, which have been suggested to be involved in the pathophysiology of fatty liver diseases [52]. In the hepatocytes, the aberrant accumulation of LDs is the hallmark of steatosis [52,53,54]. Structure, chemical composition, and tissue distribution of LDs are dynamic, which are regulated by several proteins coating on the surface of LDs [52,53,55]. These proteins include perilipins, structural proteins, lipogenic enzymes, lipases, and membrane-trafficking proteins that have been suggested to be involved in the pathophysiology of fatty liver diseases [52,53,55,56]. Five main perilipins (perilipins 1–5) are expressed in hepatocytes. Perilipin 1 surrounds mostly large LDs and was found to be upregulated in steatohepatitis due to NAFLD in humans [52,57,58]. Perilipin 2 promotes triacylglycerols accumulation, inhibits fatty acid oxidation, and impairs glucose tolerance [52]. Among these five perilipins, perilipin 2 appears to be a more reliable marker of hepatic LDs and is most upregulated in the livers of rodents and humans with NAFLD [52,57]. Total and hepatocyte-specific knock-out of perilipin 2 reduced hepatic steatosis in mice fed an obesogenic Western-style diet or a high-fat diet [52,53]. Perilipin 5 was also found to be increased in the livers of fatty liver dystrophic mice and perilipin 5 knock-out mice were protected from hepatic steatosis [52,59]. The cell death-inducing DFF45-like effector (CIDE) protein family, including CIDEA, CIDEB, and CIDEC (also known as fat-specific protein 27 (Fsp27) in mouse), are another category of LD-associated proteins [60]. Among them, CIDEA and CIDEC appear to have roles in fatty liver [52,60]. Overexpression of CIDEA and Fsp27 in mouse liver resulted in augmented hepatic lipid accumulation and the formation of large LDs [52,60]. In contrast, knockdown of these two proteins in obese mice ameliorated hepatic steatosis [52,60]. In our study, we found protein levels of perilipin 1, 2, and 5 and Fsp27 were significantly increased in livers of the HF fed mice. Except perilipin 1, protein expression of perilipin 2, perilipin 5, and Fsp27 was downregulated by 36M treatment. These data indicate that 36M prevented hepatic steatosis not only by inhibiting de novo lipogenesis but also by regulating lipid droplet proteins.

PTP1B has been found to regulate cell growth, apoptosis, differentiation, and cell movement trough dephosphorylation of the target molecules [30,61]. PTP1B is widely expressed in insulin sensitive tissues and in tissues that are affected by diabetes complications [30,62,63]. After insulin injection, PTP1B knock-out mice exhibited enhanced insulin sensitivity by increase in phosphorylation of the insulin receptor in liver and muscle tissues [64]. After being fed with the HF diet, the PTP1B^−/−^ mice were resistant to weight gain and remained insulin sensitive, while the PTP1B^+/+^ mice rapidly gained weight and became insulin resistant [64]. PTP1B also participates in the development of the liver fibrosis and other metabolic disorders [44]. Deletion of the PTP1B gene could decrease fibrosis and inhibit collagen deposition by suppressing the expression of α-smooth muscle actin and collagen 1, suggesting that inhibition of PTP1B may prevent progression of hepatic steatosis to liver fibrosis [44]. Taken together, PTP1B is considered as a negative regulator of insulin receptor signaling and PTP1B inhibitors become drug targets for metabolic diseases, such as diabetes and NAFLD. In our findings, 36M treatment reduced overexpression of PTP1B in liver and skeletal muscle of the HF-fed mice, indicating that 36M also could act as a PTP1B inhibitor.

The limitation of our study is that we did not use AMPK inhibitor to block the 36M effect on AMPK activation for further confirmation of the mechanism. In addition, whether 36M could promote translocation of GLUT4 onto cell membrane of C2C12 myotubes and skeletal muscle remains to be investigated. Another limitation is that we did not observe whether 36M could inhibit inflammatory factors such as tumor necrosis factor-alpha (TNF-α) and interleukin-6 (IL-6). Inflammation and adipocyte-derived factors including TNF-α and IL-6 have been reported to link obesity to T2DM [6,7]. Liu et al. reported that CAPE could ameliorate calcification by inhibiting activation of the NLRP3 inflammasome in human aortic valve interstitial cells [65]. KS370G (36) inhibited renal fibrosis via reduction of inflammation and oxidative stress [66]. Thus, 36M may potentially have anti-inflammation bioactivity. Whether 36M prevents the progression of T2DM and hepatic steatosis via inhibition of inflammatory factors remains to be investigated.

## 4. Materials and Methods

### 4.1. Reagents

D-glucose, 3-isobutyl-1-methylxanthine, dexamethasone, insulin, ethanol (EtOH), and dimethyl sulfoxide (DMSO) were purchased from Sigma-Aldrich Inc. (Saint Louis, MO, USA). Normal glucose (i.e., 5.5 mM) Dulbecco’s Modified Eagle Medium (DMEM; cat. 12320) penicillin/streptomycin (P/S; cat. 15140122), fetal bovine serum (FBS; cat. 10081148), and horse serum (cat. 16050122) were bought from Thermo Fisher Scientific, Inc. (Waltham, MA, USA). The purchase information for the antibodies is listed in Appendix A.

### 4.2. Preparation of N-Octyl Caffeamide

The compound 36 (KS370G) was produced according to the previous report [22]. *N*-octyl caffeamide (36M) was produced according to the following methods. First, 295 mg (1.2 equiv) of benzotriazol-1-yloxytris(dimethylamino)phosphonium hexafluorophosphate (BOP) was dissolved in 5 mL of in dichloromethane (CH_2_Cl_2_) and then the solution was added to a mixture that contained caffeic acid (100 mg; 0.56 mmol), C_8_H_17_NH_2_ (1.2 equiv) and triethylamine (Et3N; 0.08 mL) in dimethylformamide (DMF; 1.0 mL). Second, the mixture was stirred at 0 °C for 30 min, and then stirred at room temperature for 2 h. This reaction mixture was evaporated under vacuum, and the residue was partitioned between ethyl acetate (AcOEt) and H_2_O. Successively, the AcOEt layer was washed with 3 N aqueous HCl and 10% NaHCO_3_ (aq), dried over MgSO_4_ and concentrated in a vacuum. The residue was further purified by column chromatography with eluting solution (CH_2_Cl_2_-AcOEt 1:1, *v*/*v*) on silica gel (70–230 and 230–400 mesh, Merck 7734). The final products (82–88% yield) were recrystallized from AcOEt to obtain pure crystals. ^1^H and ^13^C NMR spectra were recorded on a Bruker Avance 500 spectrometer. Electron impact mass spectra (EIMS) were determined on a Finnigan TSQ-46C mass spectrometer. IR spectra were recorded on a NicoletMagna-IR 550 spectrophotometer.

36M: a colorless crystal; IR ν_max_ (cm^−1^): 3286, 1642, 1588, 1520, 1363, 1277, 1112, 975, 811; ^1^H NMR (acetone-*d*_6_, 500 MHz): *δ*_H_ 0.84 (3H, t, *J* = 6.6 Hz), 1.24 (10H, m), 1.52 (2H, quin, *J* = 6.6 Hz), 3.30 (2H, q, *J* = 6.6 Hz), 6.47, 7.42 (each 1H, d, *J* = 15.6 Hz), 6.82 (1H, d, *J* = 8.2 Hz), 6.90 (1H, dd, *J* = 8.2, 1.8 Hz), 7.09 (1H, d, *J* = 1.8 Hz); EI-MS *m/z*: 291 [M^+^]; UV (MeOH) λ_max_ (logε): 322 (4.32), 294 (4.26), 238 (3.92), 219 (4.37) nm.

The chemical structures of 36 and 36M were demonstrated in Figure 1A.

### 4.3. Cell Culture and Differentiation

Concerning the animal welfare and to reduce the use of animals, we established non-animal alternative methods to pre-screen the anti-diabetic activity and possible molecular mechanisms for natural products and novel compounds [17,67,68]. Adipose tissue and skeletal muscle play major roles in regulating whole body glucose homeostasis. Thus, we used fully differentiated mouse 3T3-L1 adipocytes and C2C12 myotubes as in vitro models. 3T3-L1 pre-adipocytes (BCRC#60159) and C2C12 myoblasts (BCRC#60083) were purchased from the Bioresource Collection and Research Center (BCRC, Food Industry Research and Development Institute, Hsinchu, Taiwan). The cells were seeded and maintained in normal glucose DMEM with 10% (*v*/*v*) FBS and 1% P/S in a humidified atmosphere of 95% air and 5% CO_2_ at 37 °C. After reaching 100% confluence, the cells were kept under incubation for an additional 48 h. For differentiation, 3T3-L1 pre-adipocytes were cultured with differentiation medium (DMEM containing 10% FBS, 450 mg/dl D-glucose, 0.32 μM insulin, 0.5 mM 3-isobutyl-1-methylxanthine, and 1 μM dexamethasone) for 2 days and then maintained in DMEM containing 10% FBS, 450 mg/dl D-glucose, and 0.32 μM insulin for another 4 days. Similarly, C2C12 myoblasts were cultured in differentiation medium (DMEM containing 450 mg/dl D-glucose and 2% horse serum) for 6 days, and the differentiated cells formed myotubes.

### 4.4. Detection of Glucose Consumption

When tested for antidiabetic activity, the culture medium of 3T3-L1 adipocytes or C2C12 myotubes was changed to DMEM containing 450 mg/dL D-glucose with or without the administration of the tested compounds. After 1, 2, 8, or 24 h, the glucose concentration in the medium was determined using a Roche Cobas Integra 400 Chemistry Analyzer (Roche Diagnostics, Taipei, Taiwan). The glucose-lowering activity was defined by calculating the concentration of glucose consumed from the culture medium. The cells without the tested compounds were assigned as control groups. Insulin (0.32 μM) and rosiglitazone were used as positive controls to confirm whether the in vitro models were sufficient to measure the glucose-lowering activity. Rosiglitazone and compounds 36 and 36M were dissolved in DMSO to make 50 μg/μL stock solutions, and then further diluted in DMSO for experimentation. The final concentration of DMSO in the medium was 0.1%.

### 4.5. Animal Studies

Six-week-old male C57BL/6J mice were obtained from the National Laboratory Animal Center (Taipei, Taiwan). The mice were caged in an air-conditioned animal facility at 22 ± 1 °C and 50–70% humidity on a 12-h light/dark cycle and were maintained with free access to water and food. They were fed either a normal chow diet consisting of 11% fat (as a percentage of total kcal), 65% carbohydrate, and 24% protein (Maintenance diet 1320, Altromin Spezialfutter GmbH & Co. KG, Lage, Germany), or a high-fat diet (HF) consisting of 45% fat, 35% carbohydrate, and 20% protein (D12451, Research Diets, Inc., New Brunswick, NJ, USA). Seven days post-arrival, mice were randomly divided into four groups: (i) normal-chow-diet-fed control mice (Con); (ii) HF-fed mice (HF); (iii) HF-fed mice with 10 mg/kg/day of 36M treatment (HF + 36M); and (iv) HF-fed mice with 10 mg/kg/day of pioglitazone treatment (HF + Pio; a positive control). There were 8 mice in each group. The administration of 36M and pioglitazone via gastric gavage was begun after 2 (prevention model) or 8 (treatment model) weeks of HF feeding. In the prevention model (Figure 3A), 36M was administered before the elevation of fasting blood glucose since the mice were fed with the HF diet for only 2 weeks. In the treatment model (Figure 4A), 36M was administered after development of T2DM. According to our previous studies, the mice had already developed obesity, insulin resistance, and high fasting blood glucose after being fed with the HF diet for 8 weeks [67,68]. Compound 36M and pioglitazone were dissolved in DMSO to make 30 μg/mL stock solutions and suspended in normal saline to make 20-fold dilutions. Depending on the body weight of the mice, the final volume administered to the mice was between 200 and 300 μL. The quantity of DMSO that the mice ingested was less than 20 μg/day, which is much lower than the US-FDA-suggested permitted daily exposure (PDE) value 50 mg/day. Compound 36M and pioglitazone were administered 5 days a week (Monday to Friday). The dose of 10 mg/kg/day administered to the mice was determined according to the previous studies of CAPE [22,27] and human recommended maximum dose (45 mg/day) of pioglitazone by multiplying 12.3 (mouse converting factor [69]) and then dividing by 60 kg (45 × 12.3/60 = 9.225 mg/kg/day; approximate to 10 mg/kg/day).

Throughout the experiment, body weight and blood glucose were monitored every two weeks. Blood glucose from the tail tip was detected by an ACCU-CHEK blood glucose meter (Roche Diagnostics, Taipei, Taiwan). The mice were sacrificed after administering the compounds for 10 weeks. After fasting overnight, the mice were anesthetized by intraperitoneal injection with Zoletil (160 mg/kg) (Virbac, Carros, France). Blood samples were collected from the hearts at the time of sacrifice, centrifuged at 3000 rpm for 15 min, and then the supernatant plasma was stored in a −20 °C freezer. A Roche Cobas Integra 400 Chemistry Analyzer (Roche Diagnostics, Taipei, Taiwan) was applied to obtain the biochemistry data, including triglyceride, total cholesterol, and alanine aminotransferase (ALT).

The liver and the skeletal muscle were collected and fixed in 10% formalin or preserved in −80 °C freezer for observing morphology by hematoxylin and eosin (H&E) staining or protein expression by Western blotting, respectively.

### 4.6. Oral Glucose Tolerance Test

After administered the compounds to the mice for 10 weeks, we performed an oral glucose tolerance test (OGTT). The mice were starved for overnight, and then the basal (0 min) blood glucose concentrations were measured before the administration of 2 g/kg glucose via gastric gavage. The following blood samples from the tail tips were taken at 15, 30, 60, and 120 min. Glucose concentration was measured using an ACCU-CHEK blood glucose meter, and area under curve (AUC) of OGTT was calculated by GraphPad Prism 9 software.

### 4.7. Protein Extraction and Western Blotting

Sodium dodecyl sulfate-polyacrylamide gel electrophoresis (SDS-PAGE) and Western blotting were used to analyze the expression levels of the target proteins. Briefly, to analyze the expression of the target proteins, the mouse liver and skeletal muscle were homogenized and lysed in M-PER Mammalian Protein Extraction Reagent supplemented with cOmplete™ Protease Inhibitor Cocktail (cat. 11697498001; Roche Applied Science). The protein concentrations were measured using a Pierce™ BCA Protein Assay kit (cat. 23227; Thermo Fisher Scientific, Inc.). For analysis of target proteins, equal amounts of tissue lysates were loaded and separated on 7.5%, 10%, or 15% sodium dodecyl sulfate (SDS)-polyacrylamide gels. After transfer to polyvinylidene difluoride (PVDF) membranes, the proteins of interest were detected using corresponding antibodies. The purchase information for the antibodies is listed in Appendix A. The blot images were quantified by ImageJ software (a public-domain Java image processing and analysis program developed at the National Institutes of Health, USA; https://imagej.nih.gov/ij/, accessed on 28 February 2019).

### 4.8. Statistical Analysis

The results for all groups are presented as mean ± standard error of the mean (SE) and were analyzed with GraphPad Prism 9 software (GraphPad Software Inc., San Diego, CA, USA). One-way analysis of variance (ANOVA) followed by Tukey’s multiple comparison test was used to analyze the differences across groups. All *p*-values were two-sided with significance accepted at <0.05.

## 5. Conclusions

In the HF-fed mice, *N*-octyl caffeamide (36M) treatment was demonstrated to lessen body weight gain and to prevent the progression of hyperglycemia and hepatic steatosis. The molecular mechanisms of 36M to treat T2DM and fatty liver are via activation of AMPK and inhibition of PTP1B. Furthermore, 36M could promote GLUT4 overexpression in skeletal muscle and inhibit lipid droplets proteins in the liver of the obese mice. Taken together, our findings suggest that 36M is a candidate compound for developing medicines against diabetes and hepatic steatosis.

## Figures and Tables

**Figure 1 ijms-23-08948-f001:**
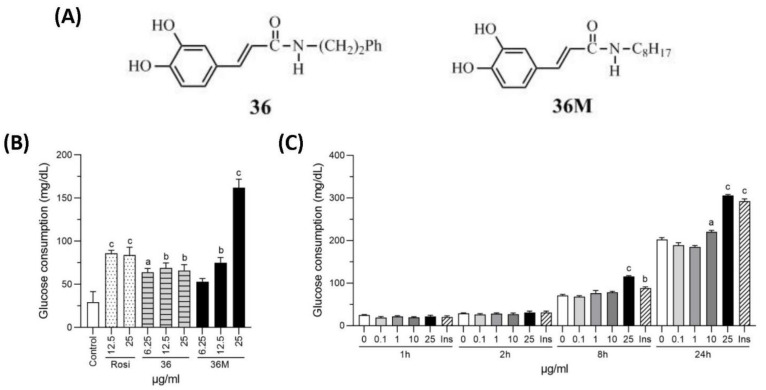
Chemical structure and glucose consumption effect of 36M. (**A**) Chemical structure of 36 and 36M. (**B**) Effect of culture medium glucose consumption in 3T3-L1 adipocytes. The cells were treated with 6.25, 12.5, or 25 μg/mL of rosiglitazone (Rosi), 36 or 36M for 24 h. (**C**) Effect of culture medium glucose consumption in C2C12 myotubes. The cells were stimulated with 0, 0.1, 1, 10, or 25 μg/mL of 36M or 0.32 μM of insulin (Ins) for 1, 2, 8, and 24 h. Data are mean ± SE (n = 3). Compared to control or 0; a: *p* < 0.05; b: *p* < 0.01; c: *p* < 0.001.

**Figure 2 ijms-23-08948-f002:**
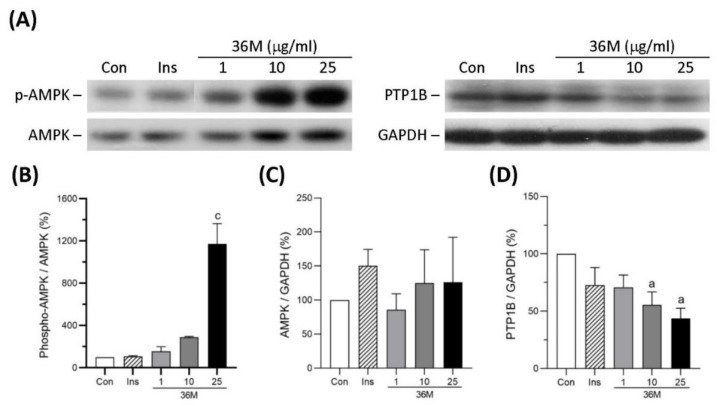
Protein expression of AMPK and PTP1B in C2C12 myotubes. (**A**) The representative immunoblot images for phosphorylated AMPK (p-AMPK), AMPK, PTP1B, and GAPDH. These protein expressions were detected by Western blotting. (**B**–**D**) The quantified results of the blots. The cells were stimulated with 1, 10, or 25 μg/mL of 36M or 0.32 μM of insulin (Ins) for 8 h. Con: control cells without treatment. Data are mean ± SE (n = 3). Compared to control; a: *p* < 0.05; c: *p* < 0.001.

**Figure 3 ijms-23-08948-f003:**
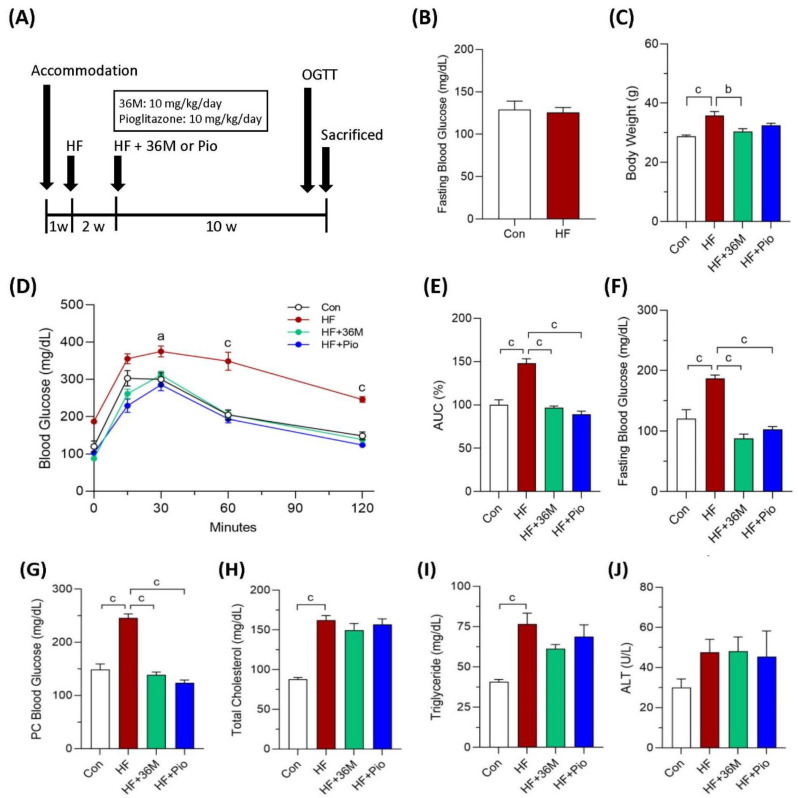
The biochemical results of the mice in the prevention model. (**A**) Scheme of the prevention model. (**B**) Fasting blood glucose of the mice after two weeks of HF feeding. (**C**) Body weight of the mice at the end point of the experiment. (**D**) Results of the OGTT of the mice detected at the 10th week of administration of 36M or pioglitazone. (**E**) Area under curve (AUC) of the OGTT results. (**F**–**J**) Fasting blood glucose, postprandial (post cibum; PC) blood glucose, total cholesterol, triglyceride, and alanine aminotransferase (ALT) of the mice at the end point of the experiment. Con: control; HF: high-fat diet; Pio: pioglitazone. Data are mean ± SE (n = 8). a: *p* < 0.05; b: *p* < 0.01; c: *p* < 0.001.

**Figure 4 ijms-23-08948-f004:**
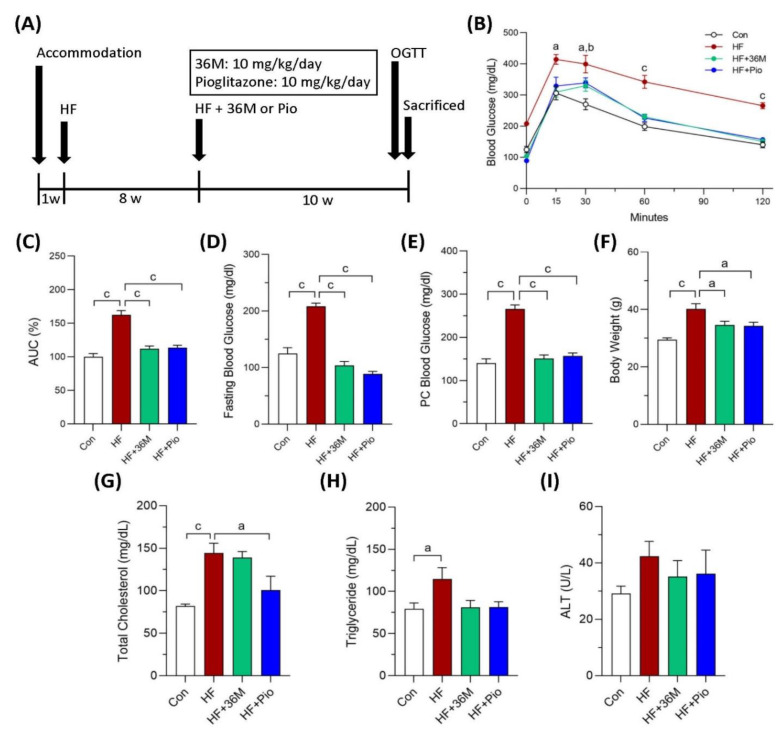
The biochemical results of the mice in the treatment model. (**A**) Scheme of the treatment model. (**B**) Results of the OGTT of the mice detected at the 10th week of administration of 36M or pioglitazone. (**C**) Area under curve (AUC) of the OGTT results. (**D**–**I**) Fasting blood glucose, postprandial (post cibum; PC) blood glucose, body weight, total cholesterol, triglyceride, and alanine aminotransferase (ALT) of the mice at the end point of the experiment. Con: control; HF: high-fat diet; Pio: pioglitazone. Data are mean ± SE (n = 8). a: *p* < 0.05; b: *p* < 0.01; c: *p* < 0.001.

**Figure 5 ijms-23-08948-f005:**
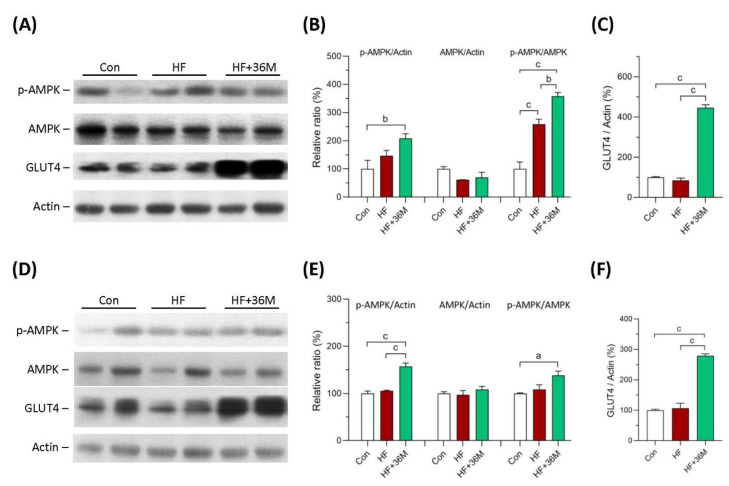
Protein expression of AMPK and GLUT4 in skeletal muscle of the mice. (**A**) The representative immunoblot images for p-AMPK, AMPK, GLUT4, and β-actin proteins of the mice in the prevention model. (**B**,**C**) The quantified results of the blots. (**D**) The representative immunoblot images for p-AMPK, AMPK, GLUT4, and β-actin proteins of the mice in the treatment model. (**E**,**F**) The quantified results of the blots. These protein expressions were detected by Western blotting. Con: control; HF: high-fat diet. Data are mean ± SE (n = 8). a: *p* < 0.05; b: *p* < 0.01; c: *p* < 0.001.

**Figure 6 ijms-23-08948-f006:**
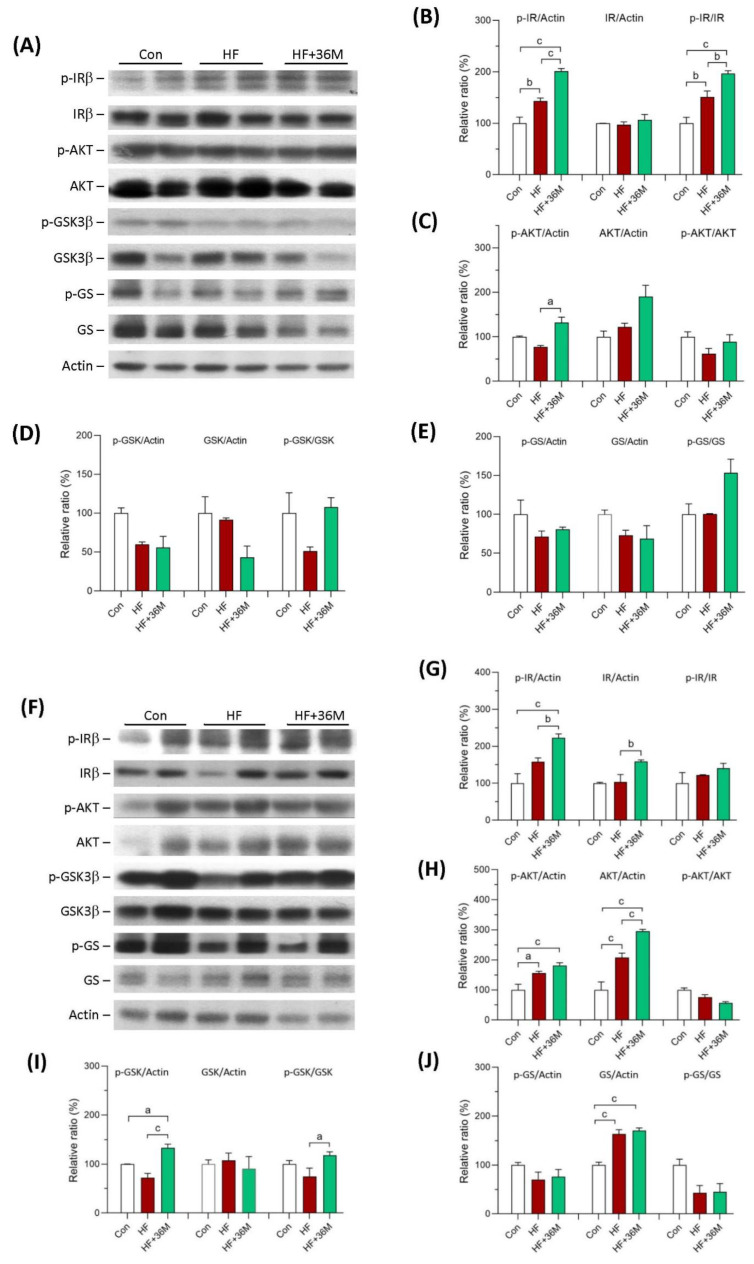
Protein expression of IRβ, AKT, GSK3β, and GS in skeletal muscle of the mice. (**A**) The representative immunoblot images for phosphorylated insulin receptor β subunit (p-IRβ), IRβ, phosphorylated AKT (p-AKT), AKT, phosphorylated GSK3β (p-GSK3β), GSK3β, phosphorylated glycogen synthase (p-GS), GS, and β-actin proteins of the mice in the prevention model. (**B**–**E**) The quantified results of the blots. (**F**) The representative immunoblot images for p-IRβ, IRβ, p-AKT, AKT, p-GSK3β, GSK3β, p-GS, GS, and β-actin proteins of the mice in the treatment model. (**G**–**J**) The quantified results of the blots. These protein expressions were detected by Western blotting. Con: control; HF: high-fat diet. Data are mean ± SE (n = 8). a: *p* < 0.05; b: *p* < 0.01; c: *p* < 0.001.

**Figure 7 ijms-23-08948-f007:**
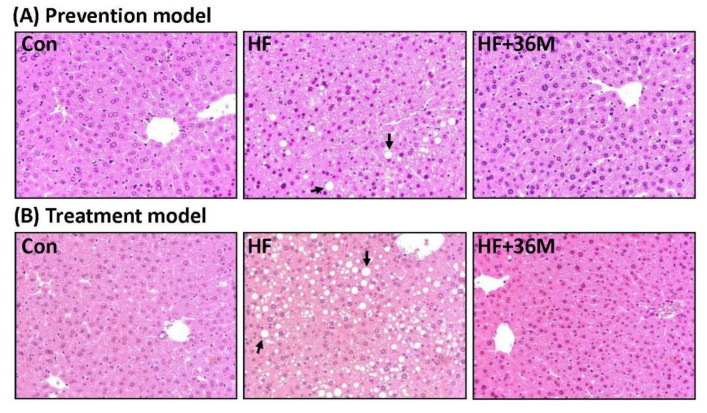
Liver morphology of the mice. (**A**) The representative images for liver hematoxylin and eosin (H&E) staining of the mice in the prevention model. (**B**) The representative images for liver H&E staining of the mice in the treatment. → indicates the representative lipid droplets. Con: control; HF: high-fat diet.

**Figure 8 ijms-23-08948-f008:**
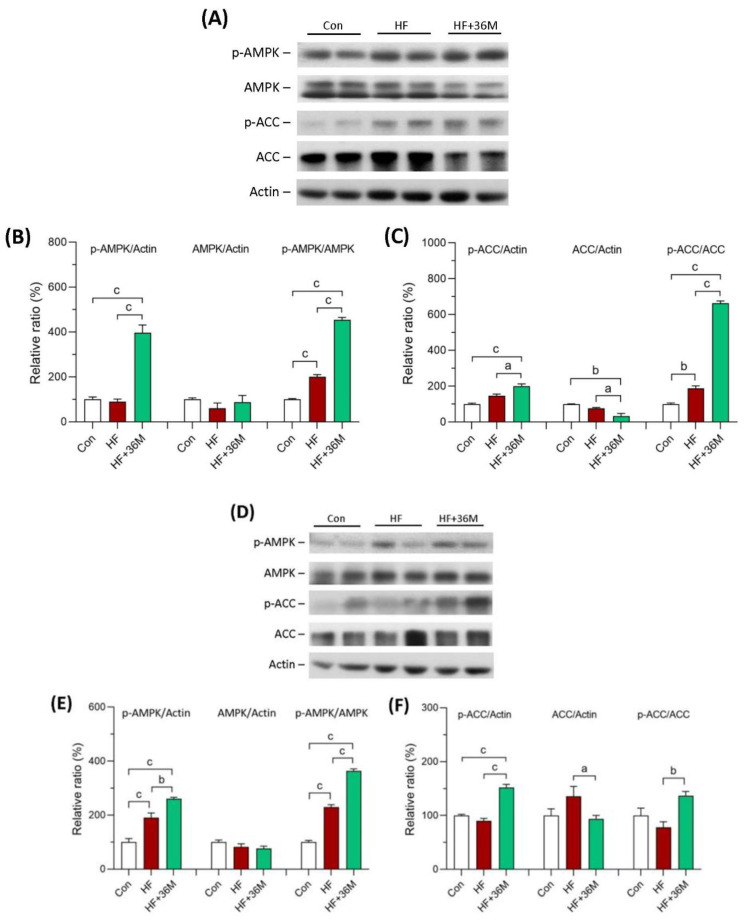
AMPK protein expression in liver of the mice. (**A**) The representative immunoblot images for p-AMPK, AMPK, p-ACC, ACC, and β-actin proteins of the mice in the prevention model. (**B**,**C**) The quantified results of the blots. (**D**) The representative immunoblot images for p-AMPK, AMPK, p-ACC, ACC, and β-actin proteins of the mice in the treatment model. (**E**,**F**) The quantified results of the blots. These protein expressions were detected by Western blotting. Con: control; HF: high-fat diet. Data are mean ± SE (n = 8). a: *p* < 0.05; b: *p* < 0.01; c: *p* < 0.001.

**Figure 9 ijms-23-08948-f009:**
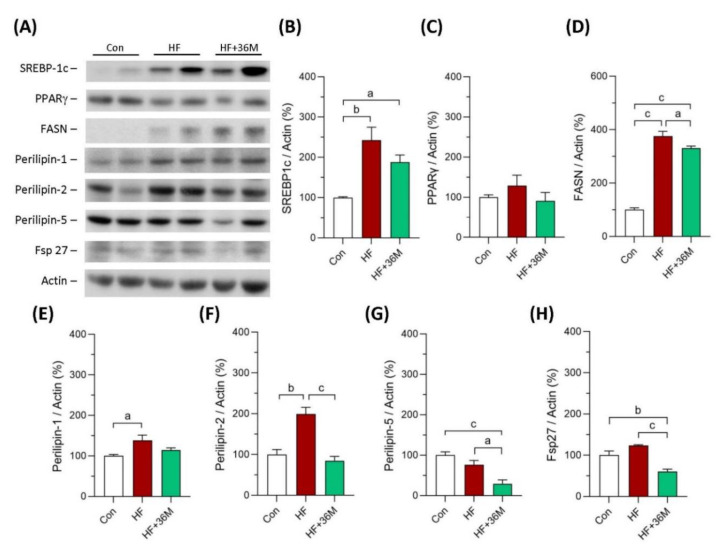
Expression of the proteins involving in lipogenesis in liver of the mice in the prevention model. (**A**) The representative immunoblot images for SREBP-1c, PPARγ, FASN, perilipin 1, perilipin 2, perlipin 5, Fsp27, and β-actin. These protein expressions were detected by Western blotting. (**B**–**H**) The quantified results of the blots. Con: control; HF: high-fat diet. Data are mean ± SE (n = 8). a: *p* < 0.05; b: *p* < 0.01; c: *p* < 0.001.

**Figure 10 ijms-23-08948-f010:**
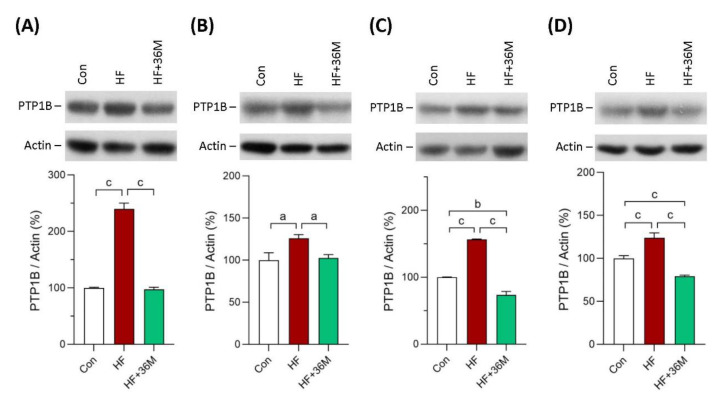
Protein expression of PTP1B. (**A**) PTP1B expression in skeletal muscle of the mice in the prevention model. (**B**) PTP1B expression in skeletal muscle of the mice in the treatment model. (**C**) PTP1B expression in liver of the mice in the prevention model. (**D**) PTP1B expression in liver of the mice in the treatment. Bar charts are the quantified results of the immunoblots. These protein expressions were detected by Western blotting. Con: control; HF: high-fat diet. Data are mean ± SE (n = 8). a: *p* < 0.05; b: *p* < 0.01; c: *p* < 0.001.

## Data Availability

Not applicable.

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
