# Peer review of "N-Octyl Caffeamide, a Caffeic Acid Amide Derivative, Prevents Progression of Diabetes and Hepatic Steatosis in High-Fat Diet Induced Obese Mice"

_ijms, 2022, doi:10.3390/ijms23168948_

Round 1

Reviewer 1 Report

Many thanks for your manuscript about the potential candidate compound N-octyl caffeamide as an AMPK inhibitor and PTP1B inhibitor for treating diabetes and hepatic steatosis. Your data are interesting and add to the sum of the literature in this field. This significant improvement in the discusión and conclusion, and the explanations of the following points are to be considered suitable for publication in a scientific journal.

Major considerations:

1)                  The title: is it really a model of diabetes? While it is true that obesity can lead to type 2 diabetes, in this case the model used is not exactly an animal model of diabetes. Below I suggest some questions about this.

2)                  In the “results” section there should be no conclusions, opinions or speculations about the study, only the description of the results. Those reflections should be in the “discussion” and/or “conclusions” sections. This can be seen for example in section 2.1, line 91; section 2.3 line 166; and section 2.6, line 408. Please modify them and review the rest of the "results" section.

3)                  In the discussion you mention " antidiabetic activity”. This term is too broad and would include both type 1 and type 2 diabetes. on the other hand, is HF-fed mice really a model of diabetes? Or insulin resistance? What is this based on? There are validated models such as Alloxan-induced diabetes, Streptozotocin-Induced Diabetic or high fat/fructose diet-induced diabetic mouse models, among others. Please describe why HF-induced obese mice would correspond to an animal model of diabetes and what type of diabetes.

  Minor considerations:

The “n” of the animals in the main text are missing.

Please clarify what you call the “prevention model”. It appears throughout the text in the results but is not explained in the main text.

Line 759: “The administration of 36M and 759 pioglitazone via gastric gavage was begun after two (prevention model) or eight (treat-760 ment model) weeks of HF feeding. Pioglitazone and compound 36M were dissolved in 761 DMSO and suspended in normal saline”. Clarify, please explain the concentrations of DMSO used in gavage since it is a toxic product for animals.

Line 810: make it clear that the difference in body weight gain is only concerning HFD and not with controls.

Author Response

We thanks reviewer's comments. The response to comments is attached.  

Reviewer 2 Report

Manuscript titled " N-octyl caffeamide, a caffeic acid amide derivative, prevents progression of diabetes and hepatic steatosis in high-fat diet induced obese mice" is a very interesting original article in the field of diabetology and cardiovascular risk factors. Overall structure is of good quality, methods and results are clear, figures are of good quality and clear to readers. However, authors should improve the manuscript in several parts:

1. Introduction should be improved, authors should add more data on other nutraceuticals that could improve the diabetes and glucose hoeostatis and reduces cardiovascular risk factors like steatosis, hsPCR and other biomarkers of inflammation ( i.e ascorbic acid, cite 10.3390/antiox9121182 )

2. in discussion, authors should describe curcuminoids also like control positive anti-inflammatory compound with anti-diabetic and hepatoprotective properties and actually available formulations able to improve theis oral bioavailability ( cite 10.1016/j.ijpharm.2015.08.039)

3. Please, discuss on the role of inflammasome in N-octyl caffeamide prevention of diabetes

4. Please add some comments on the putative anti-cytokine properties of N-octyl caffeamide, i.e its ability to reduce systemic interleukin 1 and 6. 

Author Response

(The authors gave the same response as above.)
